# Morphological, Structural, Thermal, Permeability, and Antimicrobial Activity of PBS and PBS/TPS Films Incorporated with Biomaster-Silver for Food Packaging Application

**DOI:** 10.3390/polym13030391

**Published:** 2021-01-27

**Authors:** Nurain Aziman, Lau Kia Kian, Mohammad Jawaid, Maimunah Sanny, Salman Alamery

**Affiliations:** 1Department of Food Science, Faculty of Food Science and Technology, Universiti Putra Malaysia, Serdang 43400, Malaysia; nurain7886@yahoo.com (N.A.); s_maimunah@upm.edu.my (M.S.); 2School of Industrial Technology, Faculty of Applied Sciences, Universiti Teknologi MARA, Kampus Kuala Pilah, Kuala Pilah 72000, Malaysia; 3Laboratory of Biocomposite Technology, Institute of Tropical Forestry and Forest Products (INTROP), Universiti Putra Malaysia, Serdang 43400, Malaysia; laukiakian@gmail.com; 4Laboratory of Food Safety and Food Security, Institute of Tropical Agriculture and Food Security, Universiti Putra Malaysia, Serdang 43400, Malaysia; 5Department of Biochemistry, College of Science, King Saud University, P.O. Box 22452, Riyadh 11451, Saudi Arabia; salamery@ksu.edu.sa

**Keywords:** poly (butylene succinate), tapioca starch, biomaster-silver, morphology, crystallinity, thermal stability, permeability, antimicrobial activity

## Abstract

The development of antimicrobial film for food packaging application had become the focus for researchers and scientists. This research aims to study the characteristics and antimicrobial activity of novel biofilms made of poly (butylene succinate) (PBS) and tapioca starch (TPS) added with 1.5% or 3% of Biomaster-silver (BM) particle. In morphological examination, the incorporation of 3% BM particle was considerably good in forming well-structured PBS film. Meanwhile, the functional groups analysis revealed the 3% BM particle was effectively interacted with PBS molecular chains. The flame retard behavior of BM metal particle also helped in enhancing the thermal stability for pure PBS and PBS/TPS films. The nucleating effect of BM particles had improved the films crystallinity. Small pore size features with high barrier property for gas permeability was obtained for BM filled PBS/TPS films. From antimicrobial analysis, the BM particles possessed antimicrobial activity against three bacteria *Staphylococcus aureus*, *Escherichia coli*, and *Salmonella Typhimurium* in which PBS/TPS 3% BM film exhibited strong antimicrobial activity against all tested bacteria, however, PBS/TPS 1.5% BM film exhibited strong antimicrobial activity against *E. coli* only. Hence, the incorporation of BM into PBS/TPS film could be a sustainable way for developing packaging films to preserve food products.

## 1. Introduction

The increased use of synthetic polymers for food packaging has become primary concern in publicity due to their non-biodegradability. This has inspired researchers to explore biodegradable films such as poly (butylene succinate) (PBS) for general food packaging or film. PBS is a biodegradable aliphatic polyester which has high flexibility, excellent impact strength, as well as chemical and thermal resistance formed by polycondensation reaction between 1, 4-butanediol and succinic acid [1,2]. Compared to the low density polyethylene (LDPE) with great flexible strength that frequently used in food packaging, PBS could be equally potential for such applications [3,4]. Also, because of the costly PBS material, PBS usually mixed with starch and natural fiber to achieve stable viable products and enhance their properties [5]. Starch-based materials have been widely considered in the food packaging sector as alternatives to plastics due to their biodegradability, abundance, potential for production at large scale, low cost, wide availability, ease of use, and hypoallergenic quality [6]. In addition, the plasticization of starch has enhanced the interfacial bonding of the polymer matrix [7]. Thermoplastic starch/PBS blends with malleated PBS possess good biodegradability, improved strength, and high water resistance, and are this expected to serve as a packing material [8]. Without any addition of chemicals into the films, tapioca starch is able to form flexible transparent film [9]. Furthermore, the better strength properties and thermal stability of PBS can be achieved by addition of modified tapioca starch into PBS polymer blends as compared to pure starch [10].

The growth of foodborne pathogens is one of the major factors for short-term of food shelf-life and ultimately leading to deterioration of food products. To resolve this issue, active packaging, is an innovative concept, which currently applied for developing antimicrobial food packaging due to their highly active antimicrobial reactivity, biocompatibility, great barrier performance, and promising storage ability in preserving foodstuff [11]. The effectiveness of antimicrobial packaging films in inhibiting foodborne-pathogens has been proven by some researchers. As reported by Zhao et al. [12], they had produced antimicrobial films of silver nanoparticles embedded soy protein isolate, which exhibited antimicrobial activity in against both Gram-positive and Gram-negative bacteria. Also, a nisin-coated antimicrobial film was fabricated by Mauriello et al. [13], that effectively inhibited *Micrococcus luteus* ATCC 10240 and bacterial flora in milk. Calatayud et al. [14] had reported a study on ethylene-vinyl alcohol copolymer films incorporated with 10%, 15%, and 20% cocoa extracts. Their results showed great antimicrobial activity against *S. aureus*, *Listeria monocytogenes*, *E. coli*, and *Salmonella enterica*. A chitosan-coated poly(lactic acid)/poly(butylene succinate) blend film made by Hongsriphan and Sanga [15] also able to present antimicrobial activity against *S. aureus* and *E. coli*. Apart from that, several studies on antimicrobial PBS or TPS packaging have also been done by previous researchers. The incorporation of thymol in PBS film was found to inhibit *S. aureus* and *E. coli* at 6 wt % and 10 wt %, respectively [16]. Carvacrol essential oil at 4 wt % incorporated into PBS was found to effectively inhibit *S. aureus*, while inhibit *E. coli* growth at 10 wt % [17]. The incorporation of 4% nano-ZnO in the PBS packaging could extend shelf-life of fresh-cut apples up to 18 days of cold storage [18]. Addition of TiO_2_ improved thermal stability of PET/PBS blends, and this thin film was active against both *S. aureus* and *E. coli* [19]. TPS/cellulose nanocrystal active nanocomposite films containing grape pomace extracts showed a strong antimicrobial activity against *S. aureus* ATCC 29213 [20]. TPS film incorporating ZnO nanoparticle showed antibacterial properties against *Salmonella* sp. with 6.97 mm of clear zone of diameter [9]. TPS/hydroxypropyl methylcellulose film incorporated with two antimicrobial agents, nisin and potassium sorbate was effective against *Listeria innocua* and *Zygosaccharomyces bailii* [21].

Recently, the commercialization of antimicrobial food packaging has grown rapidly since there are several companies commercializing their services to provide active packaging system. One of them is Biomaster^®^, the silver based antimicrobial packaging from Addmaster Limited, USA. Biomaster-silver (BM) was based on silver ion technology, in which the silver-based active ingredient can be used at every stage of the manufacturing process. It is available in the form of liquids or powders for textiles, plastics, paper and coatings, and has been known to be effective against various microorganisms. According to Jung et al. [22], silver ions binds with the bacteria and may cause *S. aureus* and *E. coli* to reach an active but nonculturable state and ultimately die. There are two bactericidal mechanisms of silver ions against *S. aureus* and *E. coli*; (1) As a reaction against the denaturation effects of silver ions, DNA molecules become condensed and lose their ability of replication [23]; (2) Silver ions induce the inactivation of the bacterial proteins by interact with thiol (sulfhydryl) groups in the protein [23,24,25,26], even though other target sites remain a possibility [27].

To date, there are have been no researchers reporting on the application of PBS or PBS/TPS containing Biomaster-silver particles as an antimicrobial packaging material. Thus, the aim of this research was to determine the effectiveness of Biomass-silver particles to inhibit three bacteria, *Staphylococcus aureus* (ATCC 6538P), *Escherichia coli* (ATCC 11229), and *Salmonella Typhimurium* (ATCC 14028) in PBS and PBS/TPS films. The produced PBS and PBS/TPS films in this work were also subjected to characterization to comprehensively study their properties of morphology, functional chemistry, thermal stability, crystallinity, porosity, and permeability. Hence, the novelty of this work focused on developing new types of antimicrobial PBS and PBS/TPS packaging films to preserve food products. 

## 2. Materials and Methods 

### 2.1. Materials and Chemicals

The PBS pellets (Density = 1.26 g/cm^3^; Melting point = 114 °C; Glass transition temperature = −32 °C) were purchased from PTT Public Co. Ltd. in Bangkok, Thailand. Tapioca starch (TPS) powder (Moisture = 11.1%; Bulk density = 0.63 g/cm^3^; Gelatanization temperature = 51 °C; Viscosity = 5.5 C_p_) was bought from PT. Starch Solution Int. in Karawang, Indonesia. Meanwhile, Biomaster-silver (BM) particles were provided by Indochine Bio Plastiques, Senai, Johor, Malaysia. The bacteria strains, *Staphylococcus aureus* (ATCC 6538P), *Escherichia coli* (ATCC 11229), and *Salmonella Typhimurium* (ATCC 14028) (Microbiologics, KWIK-STIKTM 2 pack, St. Cloud, MN, USA) were purchased from the Choice-Care Sdn. Bhd, Malaysia. The culture media including LB agar and LB broth were obtained from Oxoid, Malaysia.

### 2.2. Preparation of PBS and PBS/TPS Films

The films fabrication was conducted using melt-blown technique since this method was easy-handling to control the optimal interaction between different components for generating good structure of films. The pure PBS film was prepared using 100 wt % PBS pellets, while PBS/TPS film was prepared by mixing 40 wt % TPS powder with 60 wt % PBS pellets through melt-blown machine. For antimicrobial effect enhancement, different amounts of 1.5% and 3% BM particles (on the basis of total 100 wt % PBS or PBS/TPS), were incorporated to form PBS 1.5% BM, PBS 3% BM, PBS/TPS 1.5% BM, and PBS/TPS 3% BM films. The denotations and formulations for all produced films were shown in Table 1. 

### 2.3. Characterization of PBS and PBS/TPS Films

#### 2.3.1. Morphological Examination

The morphological properties of film samples were determined using a S3000-N scanning electron microscope (SEM) (Hitachi, Schaumburg, IL, USA) at 5 kV of accelerating voltage. In specimen preparation, those samples were immersed in liquid nitrogen, and then fractured to expose the cross-sectional surface. The fractured samples were loaded on a stub, and coated with platinum before viewing. 

#### 2.3.2. Functional Chemistry Analysis

The FTIR spectrum of each film sample was analyzed using an attenuated total reflection (ATR) equipped Nicolet 380 FTIR spectrometer (Thermo Fisher Scientific, Waltham, MA, USA). The analysis was run for 32 scans from 200–4000 cm^−1^ at 4 cm^−1^ resolutions.

#### 2.3.3. Thermal Analysis

A Q500 Thermogravimetric analyzer (TA Instruments, New Castle, DE, USA) was used to study the thermal stability of film samples. The TGA analysis was conducted under a nitrogen atmospheric condition from 25 °C to 1000 °C temperature range at a 20 °C min^−1^ heating rate. Also, the thermo-molecular characteristics were assessed using a DSC system. The film samples were heated from 25–600 °C temperature range at heating rate of 20 °C min^−1^ under nitrogen purge atmosphere.

#### 2.3.4. BET Analysis

The Brunauer-Emmett-Teller (BET) analysis was used to determine the surface area, total pore volume, and average pore size of film samples using a 3Flex Surface Characterization analyzer (Micromeritics, Norcross, GA, USA). Each film sample was degassed for 20 min at 60 °C with nitrogen purging before analysis. 

#### 2.3.5. Permeability Analysis

The water vapor permeability of each film sample was determined using a W3/330 Water Vapor Transmission Rate Test System (Labthink, Medford, MA, USA). The condition for flowing nitrogen gas was adjusted to partial pressure gradient of 50% to 75% relative humidity (RH) between the two chambers and the analysis was carried out at constant 25 °C temperature.

For oxygen permeability, it was assessed using a 8001 permeameter (Systech, Princeton, NJ, USA) at constant 25 °C temperature and 50% RH. Film samples were placed between the two chambers and the upstream chamber was maintained at 1 atm oxygen partial pressure, whereas the downstream chamber was flown with nitrogen sweep gas. The oxygen flux was recorded with an internal equipped coulometric oxygen sensor until reaching steady state with plateau value.

### 2.4. Antimicrobial Activity Determination 

For antimicrobial analysis, all PBS films were prepared in a roll. Those films were collected at three different locations from each roll, and labelled with different numbers of 1, 2, and 3. The ISO 22916 method [28] was used to determine the antimicrobial activity of six PBS films against bacteria of *Salmonella typhimurium* (ATCC 14028), *Escherichia coli* (ATCC 11229), and *Staphylococcus aureus* (ATCC 6538P). Active cultures were prepared by inoculating freshly cultured bacteria (24 h) from LB agar into LB broth, then incubated in an incubator shaker (200 rpm) for 24 h at 30 °C. The optical density (OD) for each culture was adjusted into 1–2 × 10^5^ CFU/mL (known as a start) by adding LB broth. The adjusted bacterial suspension (0.4 mL) was aliquoted onto each film sample (5 × 5 cm^2^) and overlaid with the same film (4 × 4 cm^2^) which facing downward, followed by incubation for 24 h in humidified condition at 30 °C. Each film sample was resuspended with 0.85% sodium chloride (4.6 mL). A serial dilution was conducted, and 100 μL for each dilution was pipetted to LB agar plates. Then, those agar plates were allowed to undergo incubation for 24 h at 30 °C, while gentamicin used as a positive control and no PBS film (no sample) was used as a negative control for the microbial culture. The antimicrobial activity of PBS films was determined from the microbial growth population after incubation. This assay was conducted triplicate to obtain average values. The microbial population was expressed as the log 10 number of CFU per ml of sample (log CFU/mL sample), and the total count was calculated for dishes containing 30–300 colonies using Equation (1) as
(1)$$\mathrm{Total}\;\mathrm{count}\;\left(\frac{\log\mathrm{CFU}}{\mathrm{ml}\;\mathrm{sample}}\right)=\log\left[\Sigma \frac{C}{V}\left(n1+0.1n2\right)D\right]$$
where C refers to the total of bacterial counts in all retained dishes; V refers to the volume used for sample dilution (mL); n1 refers to the number of retained dishes in first dilution; n2 refers to the number of retained dishes in second dilution; D refers to dilution factor of first dilution.

### 2.5. Statistical Analysis 

Statistical Analysis System (SAS) software (SAS Institute, Cary, NC, USA) was used to analyze all statistical evaluations of the results. The triplicates data for each analysis were subjected to ANOVA, and expressed as the Mean ± Standard deviation. Differences with *p* < 0.05 were considered statistically significant.

## 3. Results and Discussions

### 3.1. Morphology of PBS and PBS/TPS Films

Figure 1 depicted the cross-sectional morphology of film samples. Pure PBS film showed a smooth feature, revealing its relatively tough and ductile structure [29]. Meanwhile, PBS 1.5% BM film exhibited rougher surface feature comparing to pure PBS film. It was likely resulted by the poor interaction of BM particles with PBS matrix [30]. However, with increased BM loadings, an integrated film structure was presented by PBS 3% BM film, indicating that this compositional mixture was better in establishing intermolecular interaction between the two components [31]. 

For PBS/TPS film, it revealed ruptured surface morphology and this implied the incorporation of TPS component had somehow affected the structural integrity of PBS polymer [32,33]. Furthermore, thicker and softer film structure was observed for PBS/TPS 1.5% BM film. It might be attributed to the incompatibility of components mixture at this ratio leading to the expanded free volume in polymeric structure [34,35]. The further increased BM loading had apparently improved the compactness of polymeric structure with reduced thickness as shown by the PBS/TPS 3% BM film. This showcased the addition of 3% BM more able to compatibilize well between TPS and PBS components when comparing to 1.5% BM filling [36,37].

### 3.2. Structural Properties of PBS and PBS/TPS Films

The FTIR spectra of film samples are illustrated in Figure 2. The pure PBS film revealed its prominent peaks at about 2918 cm^−1^ (methylene group CH_2_ stretching), 1718 cm^−1^ (carboxyl group C=O vibration), 1332 cm^−1^ (ester group C–O vibration), and 1157 cm^−1^ (carbonyl group C=O vibration), representing the general functional groups of PBS polymer [10,38]. With the 1.5% BM filling, two significant sharp peaks were presented by PBS 1.5% BM and PBS/TPS 1.5% BM films at around 2856 cm^−1^ and 2918 cm^−1^. This implied that the PBS backbone structure was more likely to show up for both films, revealing the poor miscibility between 1.5% BM and PBS polymer [39]. Similarly, these characteristic peaks were also prominently exhibited by PBS/TPS film, probably due to the unreacted TPS fraction within PBS matrix, which in agreement with the ruptured morphology as observed in SEM examination. From reported work by Ayu et al. [10], who blended PBS with TPS component, also revealed similar peaks on their FTIR spectra. However, these peak intensities remained unobvious change for PBS 3% BM and PBS/TPS 3% BM films as similar with pure PBS film. This further evidenced the 3% BM filling was better in interacting with PBS molecules as compared to 1.5% BM and TPS fillings [32,40]. Moreover, the broad band observed at about 3372 cm^−1^ was with reduced intensity for PBS 3% BM film when comparing to PBS/TPS 3% BM film, showcasing the greater structural integration of 3% BM filling alone in PBS matrix without TPS component [31]. For TPS filled films, they presented a broader peak at about 1041 cm^−1^ along with a small shoulder band appeared at around 1258 cm^−1^, signaling the plasticization of TPS towards PBS molecular chains [29,35].

### 3.3. Thermal Properties of PBS and PBS/TPS Films

The thermal decomposition of film samples was evaluated by TGA curves and DTG curves as shown in Figure 3a,b, respectively, while the TGA analyzed data is listed in Table 2. Both PBS 1.5% BM and PBS 3% BM films showed slightly improved onset decomposition temperature (T_ON_) at 357.0 °C and 356.8 °C, respectively when compared with pure PBS film at 355.1 °C. It was possibly promoted by the presence of high flame retardant property of BM metal particles [41]. Interestingly, the observed horizontal curve had apparently arise before T_ON_ for PBS 1.5% BM film comparing to other curves probably due to the vaporization of entrapped water molecules that triggered weight gain during thermal heating. Furthermore, all TPS filled PBS films revealed tremendous decrement of onset decomposition temperature after the TPS introduction [10]. Meanwhile, they presented irregular shape of TGA curves, implying the inconsistent decomposition behavior of polymeric films [32,42]. However, the onset decomposition temperature was somehow improved for both PBS/TPS 1.5% BM and PBS/TPS 3% BM films when in comparison to PBS/TPS film, probably contributed by the crystallizing effect of BM silver particles [43]. 

As evaluated by DTG curves in Figure 3b, the initial peaks were presented at 326.9 °C, 318.0 °C, and 311.8 °C, for PBS/TPS, PBS/TPS 1.5% BM, and PBS/TPS 3% BM films, respectively. This signalled the earlier decomposition of TPS component [10]. The peak decomposition temperature (T_PD_) determined in this work was found in the range of 399–404 °C for each film sample, indicating the main decomposition behavior of PBS polymer. Also, the additional peaks had occurred at 481.1 °C for PBS/TPS film and at 483.3 °C for PBS/TPS 1.5% BM film. It showcased the thermal decomposition of PBS polymer was somehow modified by the different components incorporation. Nonetheless, both PBS 1.5% BM and PBS 3% BM films were regarded as the most thermally stable packaging films by attributing to the enhanced T_PD_ values as well as their symmetrical DTG curves as pure PBS film [30,40].

Thermo-molecular property of film samples is assessed with DSC analysis as displayed in Figure 4 and the data is summarized in Table 3. From the analyzed results, PBS 1.5% BM film showed higher crystallization temperature (T_c_), whereas PBS 3% BM film presented lower T_c_ when in comparison with pure PBS film. This indicated the polymeric solidification behavior had been better improved for PBS 3% BM film when comparing to PBS 1.5% BM film [30]. 

Additionally, the crystallization temperature for PBS/TPS film was nearly closed to pure PBS film, implying the incorporation of TPS component did not affects the crystallization behavior of PBS molecules [33,44]. Nevertheless, the introduction of BM filling had further reduced the T_c_ of PBS/TPS 1.5% BM and PBS/TPS 3% BM films. It evidenced the BM silver particles could play the role in crystallizing the PBS polymer. Apart from that, all film samples presented insignificant change of melting temperature (T_m_), possibly contributed by the large portion of PBS used this work. However, the melting enthalpy (∆H_m_), which corresponding to crystallinity (X_C_) was different for each film sample. The PBS 1.5% BM film presented decreased ∆H_m_ and X_C_ as compared to pure PBS film, showcasing lesser energy amount was needed to melt down the low crystalline polymeric structure [29]. In among all film samples, PBS 3% BM film was determined with the highest X_C_ with 37.06%, which promoted by the great nucleating effect of 3% BM filling [41,45]. Also, PBS/TPS 3% BM film is the only sample exhibited X_C_ value that closed to pure PBS film. This proved that the compositional mixture of PBS/TPS 3% BM film was optimum to maintain the crystal structure of PBS packaging film [39,46].

### 3.4. Porosity, Water Vapor, and Oxygen Gas Permeability of PBS and PBS/TPS Films

BET, water vapor, and oxygen gas permeability analysis of film samples is shown in Table 4. From BET analysis, the total pore volume and average pore diameter were increased for both PBS 1.5% BM and PBS 3% BM films when comparing to pure PBS film. It was because the poor miscibility between BM and PBS resulting in the large voids formation. However, the PBS 3% BM film showed a remarkably lower surface area than the other two films. This indicated that the 3% BM filling was well interacted with PBS polymer and subsequently formed an integrated film structure as examined by SEM. 

For PBS/TPS film, its average pore diameter was dramatically increased up to 604.90 nm, whereas its total pore volume and surface area were decreased to as low as 0.102 cm^3^/g and 0.336 m^2^/g, respectively, owing to the contribution of TPS plasticizing structure. Furthermore, the average pore diameters were tremendously reduced to 128.97 nm and 33.07 nm for PBS/TPS 1.5% BM and PBS/TPS 3% BM films, respectively. This implied that the nucleating effect of BM particles were more likely to show up when mixing with TPS and PBS components together [47,48]. The total pore volume remained insignificant change for PBS/TPS 1.5% BM and PBS/TPS 3% BM films as comparing to PBS/TPS film, possibly the TPS plasticizer still taking strong effect in enclosing the porosity of the film. Moreover, the gradual increment of surface area for PBS/TPS 1.5% BM and PBS/TPS 3% BM films probably driven by the formation of small size pores that ultimately increased the surface roughness of the films [38,49].

Water vapor and oxygen permeability of film samples are illustrated in Figure 5a,b, respectively. Pure PBS film showed relatively great barrier to water vapor permeability by attributing to its well assimilated polymeric film structure. With BM loadings, the water vapor permeability was increased for PBS 1.5% BM and PBS 3% BM films. It was attributed to the polar characteristic of BM silver ions that favored the water molecules affinity towards film [17,30]. Besides this, the water vapor permeability was significantly increased for PBS/TPS film, due to the hydrophilic nature of TPS component which facilitated the water penetration [36,44]. The further incorporation of BM had improved the water barrier for PBS/TPS 1.5% BM and PBS/TPS 3% BM films. This was promoted by the reduced pore sizes for both films through BM nucleation as mentioned in BET analysis. 

As for oxygen permeability (Figure 5b), only PBS 1.5% BM film showed poorer oxygen barrier comparing to pure PBS film, which resulted by its larger total pore volume and bigger pore size properties. However, PBS 3% BM film still presented reduced oxygen permeability probably contributed by its lower surface area formed by good polymer interaction [43,49]. Furthermore, the oxygen barrier was remarkably enhanced for PBS/TPS, PBS/TPS 1.5% BM, and PBS/TPS 3% BM films, signaling the componential mixture used in this study was promising to produce highly impermeable PBS packaging film for preserving food product. The schematic diagram below briefly revealed the mechanism of produced PBS/TPS films in response to the penetration of oxygen and water (Figure 6).

### 3.5. Antimicrobial Activity of PBS and PBS/TPS Films

The antimicrobial activity of film samples against *S. aureus* (ATCC 6538P), *E. coli* (ATCC 11229), and *S. Typhimurium* (ATCC 14028) are shown in Figure 7, Figure 8 and Figure 9, respectively. This activity was compared with the positive control (gentamicin) and negative control (no PBS film/no sample). The cell number of each culture for this study was standardized into 5 log CFU/mL and known as at start. Based on Figure 7, Figure 8 and Figure 9, the populations of three bacteria without PBS film (no sample) were increased (*p* < 0.05) after incubation from 5 (at start) to 6, 6 and 7 log CFU/mL, respectively. 

In Figure 7, the pure PBS film was not significantly different (*p* > 0.05) with the negative control (no sample), however, the PBS/TPS film exhibited slightly reduction (*p* < 0.05) of *S. aureus* (ATCC 6538P) population. In comparison between control (pure PBS and PBS/TPS films) and treated film samples, the populations of *S. aureus* (ATCC 6538P) were decreased (*p* < 0.05) by addition of BM into the films. The PBS/TPS 3% BM film exhibited strong antimicrobial activity, and this sample was comparable with positive control, gentamicin. This indicates that PBS/TPS 3% BM film showed the killing effect on *S. aureus* (ATCC 6538P). The other treated film samples also showed antimicrobial activity as follow: PBS/TPS 1.5% BM > PBS 3% BM > PBS/TPS 1.5% BM films against *S. aureus* (ATCC 6538P).

In Figure 8, the pure PBS film was not significantly different (*p* > 0.05) with the negative control (no sample), and the PBS/TPS film exhibited slightly reduction (*p* < 0.05) of *E. coli* (ATCC 11229) population. Addition of BM into the films also showed good antimicrobial activity against *E. coli* (ATCC 11229), especially the PBS/TPS 1.5% BM and PBS/TPS 3% BM films, and these films were comparable with positive control, gentamicin. The other treated film samples also showed antimicrobial activity as follow: PBS 3% BM > PBS 1.5% BM films against *E. coli* (ATCC 11229).

In Figure 9, both pure PBS and PBS/TPS films was not significantly different (*p* < 0.05) with the negative control (no sample). However, the BM only exhibited good antimicrobial activity against *S. Typhimurium* (ATCC 14028) in PBS/TPS film. The effective sample as well as gentamicin was PBS/TPS 3% BM film. The other treated film samples also showed antimicrobial activity as follow: PBS 3% BM > PBS/TPS 1.5% BM ≥ PBS 1.5% BM films against *S. Typhimurium* (ATCC 14028).

Among six PBS films, the treated PBS films (with BM) showed the lowest (*p* < 0.05) populations of all three bacteria than the pure PBS and PBS/TPS films. This indicates that the BM exhibited antimicrobial effects even in pure PBS or PBS/TPS films. The higher percentages of each BM exhibited a good antimicrobial activity compared to the lowest percentages of BM. However, in comparison among four treated PBS films (with BM), the addition of BM into the PBS/TPS film was more effective compared to the addition of BM into the pure PBS film. It was probably contributed by the presence of hydrophilic TPS component in PBS film, that helped in interacting with ionic BM through physisorption process, and while the other active sites of BM with positive charge able to continuously reacting with bacterial cell membrane, thereby suppressing their growth. Herein, the antimicrobial testing is regarded as the only analysis which could strongly prove the presence of BM in PBS film by showing off its antimicrobial effect, despite the amount of BM used is remarkably low and undetectable by other analyses in this study. 

In this study, we also found that the treated film samples were more effective against *E. coli* (ATCC 11229), followed by *S. aureus* (ATCC 6538P) and *S. Typhimurium* (ATCC 14028). The bacterial resistance was shown by the different degree of antimicrobial activity from each film in against each tested bacteria. Besides this, the superior suppression of bacterial growth presented by the produced films in our work, especially for PBS/TPS 3%BM was comparable to the work reported by Wattanawong et al. [50], who produced silver loaded zeolite filled PBS films with 99.9% of bacterial reduction on *E. coli* and *S. aureus*. Also, from the reported work by Warsiki and Bawardi [9], their tapioca starch/zinc oxide (ZnO) antimicrobial films could exhibit antimicrobial activity in against *Salmonella sp.* and *E. coli* at 1–2 wt % ZnO, however the ZnO nanoparticle filled PBS antimicrobial film produced by Petchwattana et al. [51] could only inhibit *S. aureus* and *E. coli* at above 6 wt % ZnO. Hence, the using of less amount antimicrobial agent, BM in our work was efficient in giving great antimicrobial effect, for developing high-performed antimicrobial packaging films.

## 4. Conclusions

In conclusion, the BM was successfully incorporated in PBS and PBS/TPS films using melt-blown technique. From morphology examination, the PBS 3% BM and PBS/TPS 3% BM films revealed good film structure. This was supported by the FTIR analysis showing that both film samples had insignificant change of spectra compared to pure PBS film. Meanwhile, the high flame retardancy property of BM metal particles had improved the thermal resistance for all BM filled PBS films. The crystallizing effect of BM particles also aided in enhancing the crystallization behavior of PBS/TPS film. Moreover, the good compositional mixture of PBS/TPS 1.5% BM and PBS/TPS 3% BM films endowed them with high surface area and total pore volume, as well as small pore size features. This further contributes a great barrier for water vapor and oxygen gas permeability. By antimicrobial analysis, the BM acted as an antimicrobial agent in PBS films against *S. aureus* (ATCC 6538P), *E. coli* (ATCC 11229), and *S. Typhimurium* (ATCC 14028). The PBS/TPS 3% BM film showed strong antimicrobial activity against all three bacteria as well as gentamicin, whereas the PBS/TPS 1.5% BM film only presented antimicrobial effect against *E. coli* (ATCC 11229). Hence, it is suggested that the BM filled PBS/TPS films can be used in food packaging application in the future.

## Figures and Tables

**Figure 1 polymers-13-00391-f001:**
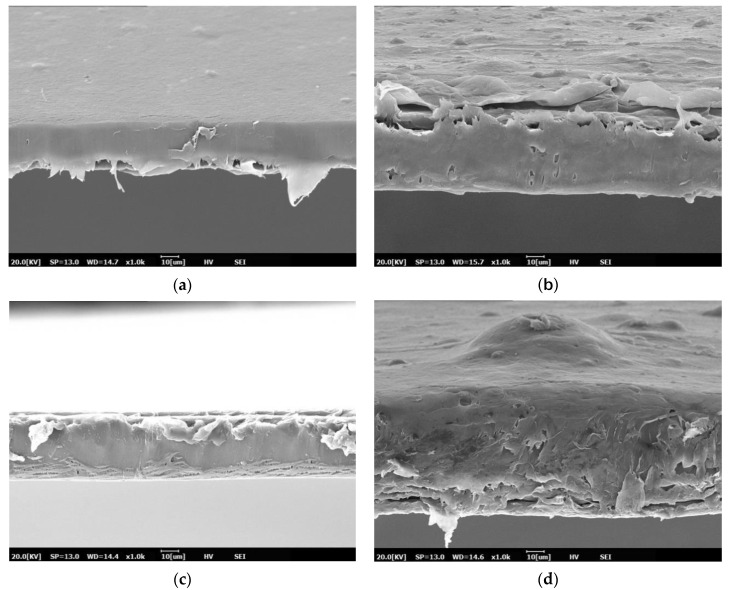
SEM images of cross-sectional morphology for film samples of (**a**) PBS, (**b**) PBS 1.5% BM, (**c**) PBS 3% BM, (**d**) PBS/TPS, (**e**) PBS/TPS 1.5% BM, and (**f**) PBS/TPS 3% BM under ×1000 magnification viewing.

**Figure 2 polymers-13-00391-f002:**
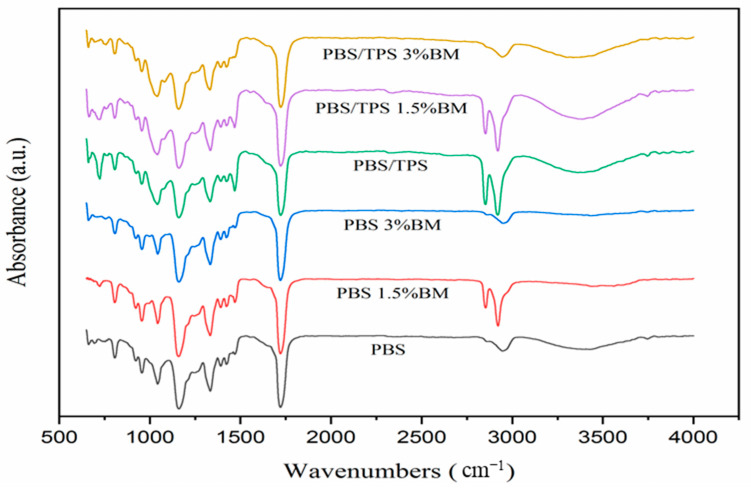
FTIR spectra of PBS films.

**Figure 3 polymers-13-00391-f003:**
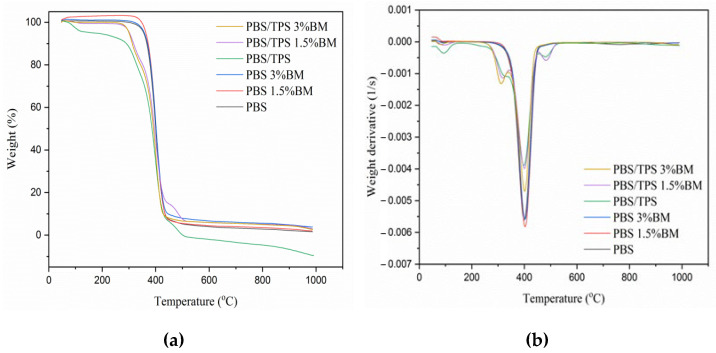
(**a**) TGA and (**b**) DTG curves of PBS films.

**Figure 4 polymers-13-00391-f004:**
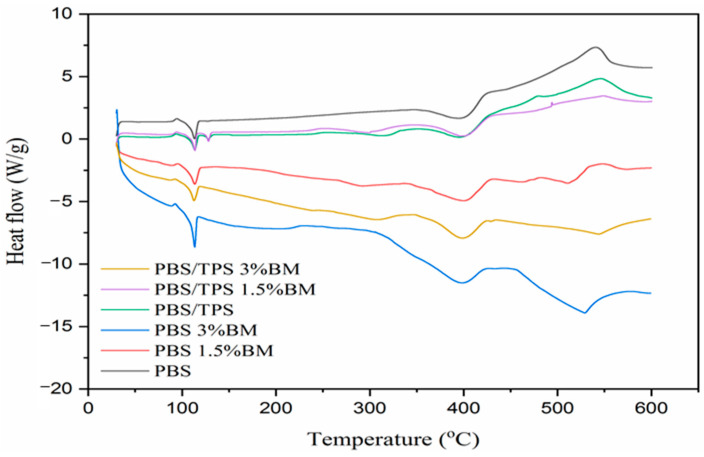
DSC spectra of PBS films.

**Figure 5 polymers-13-00391-f005:**
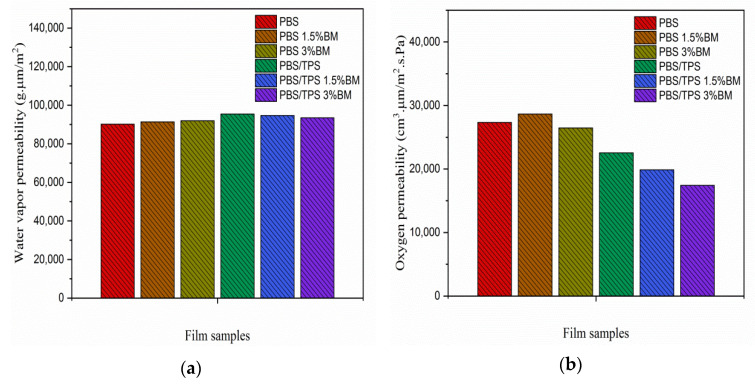
(**a**) Water vapor and (**b**) oxygen gas permeability of PBS films.

**Figure 6 polymers-13-00391-f006:**
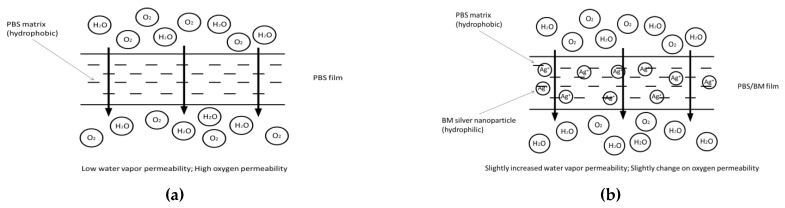
Schematic diagrams of (**a**) PBS, (**b**) PBS/BM, (**c**) PBS/TPS, and (**d**) PBS/TPS BM films in response to the penetration of oxygen and water.

**Figure 7 polymers-13-00391-f007:**
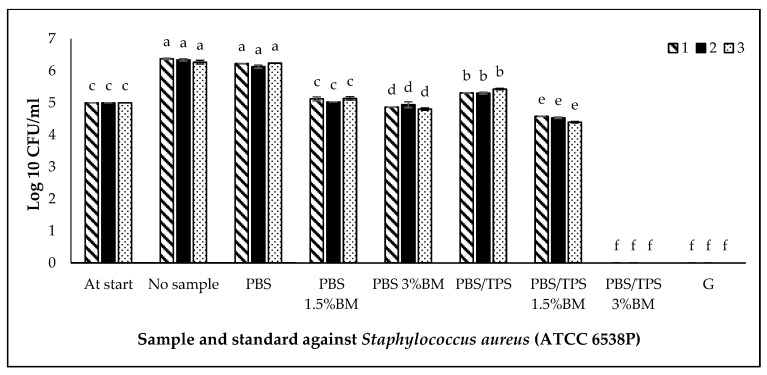
*Staphylococcus aureus* (ATCC 6538P) growth on different PBS films. Note: At start: the cell number of culture before incubation (5 log CFU/mL); No sample: no PBS film; PBS: PBS film without antimicrobial agent; PBS 1.5% BM: PBS with 1.5% BM; PBS 3% BM: PBS with 3% BM; PBS/TPS: PBS with tapioca starch; PBS/TPS 1.5% BM: PBS/tapioca starch with 1.5% BM; PBS/TPS 3% BM: PBS/tapioca starch with 3% BM; G: Gentamicin. Error bars represent standard deviation (n = 3). The different a–f small letters indicate significant difference (*p* < 0.05) among film samples.

**Figure 8 polymers-13-00391-f008:**
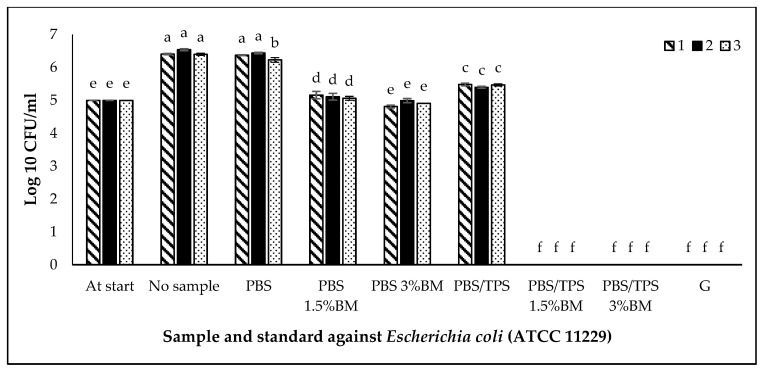
*Escherichia coli* (ATCC 11229) growth on different PBS films. Note: At start: the cell number of culture before incubation (5 log CFU/mL); No sample: no PBS film; PBS: PBS film without antimicrobial agent; PBS 1.5% BM: PBS with 1.5% BM; PBS 3% BM: PBS with 3% BM; PBS/TPS: PBS with tapioca starch; PBS/TPS 1.5% BM: PBS/tapioca starch with 1.5% BM; PBS/TPS 3% BM: PBS/tapioca starch with 3% BM; G: Gentamicin. Error bars represent standard deviation (n = 3). The different a–f small letters indicate significant difference (*p* < 0.05) among film samples.

**Figure 9 polymers-13-00391-f009:**
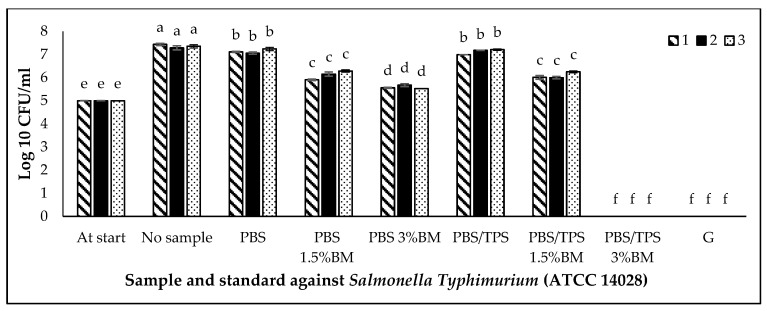
*Salmonella Typhimurium* (ATCC 14028) growth on different PBS films. Note: At start: the cell number of culture before incubation (5 log CFU/mL); No sample: no PBS film; PBS: PBS film without antimicrobial agent; PBS 1.5% BM: PBS with 1.5% BM; PBS 3% BM: PBS with 3% BM; PBS/TPS: PBS with tapioca starch; PBS/TPS 1.5% BM: PBS/tapioca starch with 1.5% BM; PBS/TPS 3% BM: PBS/tapioca starch with 3% BM; G: Gentamicin. Error bars represent standard deviation (n = 3). The different a–f small letters indicate significant difference (*p* < 0.05) among film samples.

**Table 1 polymers-13-00391-t001:** Denotations and formulations of produced films.

No.	Film Denotations	Amount of TPS (wt %)	Amount of BM (wt %)
1	PBS	N/A	N/A
2	PBS 1.5% BM	N/A	1.5
3	PBS 3% BM	N/A	3
4	PBS/TPS	40	N/A
5	PBS/TPS 1.5% BM	40	1.5
6	PBS/TPS 3% BM	40	3

Note: N/A: not available.

**Table 2 polymers-13-00391-t002:** TGA data of PBS films.

Samples	T_ON_ (°C) ^a^	T_PD_ (°C) ^b^	W_ML_ (J/g) ^c^	R_F_ (%) ^d^
PBS	355.1	399.3	95.62	4.75
PBS 1.5% BM	357.0	403.2	97.81	5.36
PBS 3% BM	356.8	402.5	93.04	7.80
PBS/TPS	283.7	399.5	96.08	1.19
PBS/TPS 1.5% BM	288.7	399.8	93.11	6.30
PBS/TPS 3% BM	290.2	401.9	93.38	6.50

Note: ^a^ Onset decomposition temperature; ^b^ Peak decomposition temperature; ^c^ Maximum weight loss; ^d^ residue formation.

**Table 3 polymers-13-00391-t003:** DSC data of PBS films

Samples	T_c_ (°C) ^a^	T_m_ (°C) ^b^	∆H_m_ (J/g) ^c^	X_C_ (%) ^d^
PBS	94.4	113.3	69.42	34.71
PBS 1.5% BM	95.3	113.7	63.53	32.25
PBS 3% BM	92.2	113.5	71.89	37.06
PBS/TPS	94.5	113.3	51.50	28.61
PBS/TPS 1.5% BM	94.0	113.6	56.53	31.94
PBS/TPS 3% BM	92.0	112.6	62.20	35.75

Note: ^a^ Crystallization temperature; ^b^ Melting temperature; ^c^ Melting enthalpy; ^d^ Crystallinity degree, X_C_ = ∆H_m_/[∆H^o^_m_ (1 − W_f_)] × 100%, where ∆H^o^_m_ is the enthalpy of 100% pure PBS (200 J/g); W_f_ is the weight fraction of active agents in the PBS film.

**Table 4 polymers-13-00391-t004:** BET, water vapor, and oxygen gas permeability data of PBS films

Samples	BET			Permeability	
	Surface Area (m^2^/g)	Total Pore Volume (cm^3^/g)	Average Pore Diameter (nm)	Water Vapor (g μm/m^2^)	Oxygen (cm^3^ μm/m^2^ s Pa)
PBS	3.598	0.211	117.21	90,170	27,320
PBS 1.5% BM	3.572	0.414	231.62	91,370	28,650
PBS 3% BM	3.030	0.436	287.44	91,940	26,470
PBS/TPS	0.336	0.102	604.90	95,400	22,540
PBS/TPS 1.5% BM	1.504	0.097	128.97	94,660	19,860
PBS/TPS 3% BM	4.872	0.081	33.07	93,500	17,420

## Data Availability

Not applicable.

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
