# Peer review of "Morphological, Structural, Thermal, Permeability, and Antimicrobial Activity of PBS and PBS/TPS Films Incorporated with Biomaster-Silver for Food Packaging Application"

_polymers, 2021, doi:10.3390/polym13030391_

Round 1
Reviewer 1 Report
This research aims to study the characteristics and antimicrobial activity of novel biofilms made of PBS and TPS added with 1.5 % or 3 % of BM particle. The produced PBS and PBS/TPS films were also subjected to characterization to comprehensively study their properties of morphology, functional chemistry, thermal stability, crystallinity, porosity, and permeability. The novelty of this work focused on developing antimicrobial PBS and PBS/TPS packaging films to preserve food products. The experiments were well designed and carefully performed, and the manuscript is well organized. Therefore, this manuscript could be considered for publication in Polymers after a minor revision. 1.Thermal analysis – please detail on the machine used. 2.Line 189: “It might be attributed to the incompatibility of those three components and subsequently leading to the expanded polymeric structure.” Line 192: “This showcased the 3 % BM filling could contribute well in compatibilizing between TPS and PBS molecules.” The above two analyses seem to conflict. 3.“1.5%BM” and “3%BM” seem to lack spaces. Please, update throughout the text. 4.Figure 3: Before thermal decomposition, the quality of PBS 1.5 % BM seems to have increased. Please analyze the reason. 5.The references seem to have a format error, for example: 14, 18, 25, 28, 39. 6.Some related references are suggested to be cited such as Journal of Applied Polymer Science, 2020, DOI: 10.1002/app.50320; Green Chem., 2020, 22, 7622-7664; ACS Omega, 2020, 5, 38, 24256-24261; Green Chem., 2019, 21, 4449–4456.
Author Response
Here attached Reply

Reviewer 2 Report
The manuscript polymers-1031012 proposes new potential antimicrobial films for food packaging, based on poly(butylene succinate) (PBS), PBS/tapioca starch (TPS) and using Biomaster-silver particles as antimicrobial agent.
The experimental study is potentially interesting, but several pieces of information are missing (see below) and the preparation protocol for obtaining homogenous PBS/BM, PBS/TPS and PBS/TPS/BM films is questionable. The current morphological observations and ATR-IR spectra are not justifying the statement of a homogeneous structure of PBS/TPS blends nor the good dispersion of BM inside the polymer matrix. These are the main aspects to be fixed, for improving the work consistency.
Please find below some other comments intended to improve the experimental report and the work consistency.
Several paragraphs and ambiguous scientific expressions need to be reformulated for the sake of scientific clarity and pertinence. For instance:
- The introductory part needs to be reorganized to present first the PBS and its potential for sustainable food packaging, then the economic interest to use PBS/TPS blends instead of PBS alone, for food packaging applications. The state-of-the art on the antimicrobial food packaging films should be reorganized according to the nature of the antimicrobial agent that has been used (nanoparticles, like Ag NPs or ZnO, essential oils such as thymol, antimicrobial polymers such as chitosan, etc.). in the present form, it is very difficult to identify the main effective tendencies to obtain antimicrobial PBS-based films for food packaging.
- Line 44: Please correct the reported antimicrobial system. “As reported in [2], the soy protein isolation/antimicrobial silver nanoparticles films”. Isolate instead of isolation. Furthermore, silver nanoparticles ARE antimicrobial per se, so no need to add “antimicrobial” in front of Ag NPs.
Secondly, it is very inelegant to cite “as reported in [2]”, without the authors name(s), especially when for the other references, the authors are mentioned in the manuscript.
- Lines 44-46: “synthetic packaging films has a major concern of disposal due to their total nonbiodegradability.“
Here, an important confusion between synthetic = nonbiodegradable vs. [which class of polymers, not clear from this manuscript (???)] = biodegradable or compostable.
Please keep in mind that poly(butylene succinate) (PBS) is an aliphatic polyester synthesized industrially from monomers obtained from fossil-based or renewable resources. PBS is biodegradable regardless its petrochemical or (more recently) renewable resources, as the polymer biodegradability does not depend on its renewable or fossil-based type of resources but on the chemical nature of the polymer chains. Poly-epsilon-caprolactone (PCL) is another biodegradable synthetic polymer. LDPE (addressed in the manuscript) is non-biodegradable, regardless the fossilbased or renewable resources from which it is produced at industrial scale.
- Lines 50-58: The comparison PBS vs. LDPE, from food packaging viewpoint, should be better organized, according to their common features vs differences. A large choice of references is available to assess the interest of PBS for food packaging in general (films and co.).
- Lines 57-58: “Due to its compostability and oil migration resistance at elevated temperatures properties,
PBS also use as a viable alternative to petroleum polymers and perfluorinated chemicals [9].”
Viable alternatives are expected to fulfill the same list of key-properties (mechanical, physico- chemical, thermal) as the generally-used polymers for food packaging applications. Please clearly explain WHY the PBS is a viable alternative to petroleum polymers and perfluorinated chemicals. Which are the intrinsic properties of the PBS allowing it to replace the petroleum polymers? Same question about the replacement of perfluorinated chemicals. Is this later example relevant for the present work?
- Lines 98-99: “The novelty of this work focused on developing antimicrobial PBS and PBS/TPS packaging films to preserve food products” - Please explicit “new types of antimicrobial PBS and PBS/TPS […]” for the sake of truth.
- Preparation of the PBS and PBS/TPS films, with and without BM agent: The manuscript indicates the use of melt-blown technique to generate the PBS and PBS/TPS films. However, the well-known meltblown process is not intended for compounding polymers or dispersing powder particles inside polymer matrices. Please explicit the processing stages and the processing conditions (temperature, type of extruder) used for producing PBS and PBS/TPS films, with and without BM agent.
- Lines 109-113: The entire paragraph needs to be reorganized and re-written, as the current formulation is ambiguous. Several PBS and PBS/TPS films where prepared, with respectively 0 – 1.5 – 3 wt% of BM. The PBS and PBS/TPS films served as references for evaluating the antimicrobial effectiveness of the antimicrobial formulations. The sentence from lines 117-118 is difficult to understand.
- Please replace the titles 2.2 - 2.3 - 3.1 - 3.2. - 3.3 - 3.4. - 3.5 from “PBS films” with “PBS and PBS/TPS films”, for sake of clarity
Several important elements are missing from the manuscript and need to be clearly presented:
- The PBS homopolymer is a highly crystalline polymer. Which is the molecular weight or the equivalent melt flow index (MFI) of the PBS grade used in this study? Which is its announced melting temperature? Announced water and oxygen permeabilities? (from technical datasheet)
- Which are the announced characteristics of the TPS used in this study, such as amylose content, moisture content, processing temperature? (from technical datasheet) Which are the typical permeability data for this kind of polymer films? (from literature data)
- Which is the PBS/TPS ratio used in this work?
- What exactly is the Biomaster grade used in this study? Are they pure Ag nanoparticles? Are these nanoparticles covered by some polymer shell, in order to facilitate incorporation into polymer matrices? Which are Biomaster characteristics – average particle size, size distribution etc.? These elements are very important for understanding how to handle with it and how to disperse it efficiently into the polymer matrix. This will also allow understanding the interface between the antimicrobial agent and PBS and PBS/TPS blends, respectively.
- In the Results and Discussion part: Which are the literature records about the compatibility, typical morphologies, mechanical and thermal properties of PBS/TPS blends with similar or close PBS/TPS ratio as in this study? Please compare the experimental results from this study with the exhaustive literature data on PBS and PBS/TPS blends, with and without antimicrobial agents.
- The experimental results reported in the section 3.1 are unclear. First, the legend of Figure 1 announces data about polymers blends which are not presented in the manuscript: PBS/TPS with 1 and 2 wt% BM, respectively. Secondly, the scalebars are so small that cannot be read. Thirdly, and most important, the magnitude used in six SEM images does not allow observing if:
- BM is well dispersed inside PBS matrix
- PBS/TPS blends have a homogeneous structure
- BM is well dispersed inside PBS/TPS blends
Considering the reported preparation stage, were the key-compounding stage is missing and the dispersion stage inside a simple melt-blown extruder is not sufficient to allow a good dispersion of the TPS inside PBS and of the BM inside the polymer matrix. At this stage of the work, clear proofs of the homogeneous morphologies of PBS and PBS/TPS blends are required, with clear evidence of good dispersion. – In case the samples are not homogeneous, the entire experimental work risk to be compromised.
- The ATR-IR spectroscopy has a limited path length into the sample. Consequently, inhomogeneous samples – such as PBS/TPS heterogeneous blends and antimicrobial heterogeneous formulations – risk to give inconsistent spectra, difficult to explain. The inconsistencies one can see in Figure 2 might be caused by the fact that the preparation stage, more explicitly the compounding stage is missing or not efficient to allow a good dispersion of the TPS inside PBS and of the BM in the polymer matrix. The results and discussion on ATR-IR results should be completely re-written, but most important, the morphology and ATR-IR analyses should be repeated and sent in an additional file, in order validate the luck of coherence reported in Figure 2 (the presence of TPS and BM, respectively), despite a supposed homogeneous structure of all the films. Please also add an ATR-IR spectrum of TPS alone.
In absence of clear proofs of the homogeneous morphologies of PBS and PBS/TPS blends, with clear evidence of good dispersion, interfaces etc., the entire experimental work risk to be compromised.
The experimental protocols for evaluating the water and oxygen permeabilities of the different films, and the antimicrobial effectiveness of BM in the reported doses (1.5 and 3.0 wt%, respectively) are correct.
However, the experimental results highly depend on the effective dispersion of the antimicrobial agent inside PBS and PBS/TPS films.
At this stage of the work, it is difficult to understand, and the manuscript does not clearly justify: - the luck of effectiveness of 3% of BM in PBS, while the same 3% of BM in PBS/TPS manages to completely suppress the different microbial cultures (see Figures 6-7-8).
- the difference in film permeabilities – as reported in Figure 5. Please propose an additional figure with schematic pathways of water and oxygen molecules (respectively) through the film cross-section, in order to explain the differences between PBS alone, PBS/BM, PBS/TPS blends and PBS/TPS with BM.
The entire work needs also systematic comparison with literature data on PBS and or PBS/TPS blends with silver nanoparticles.
In conclusion, the manuscript requires major revision before it can be considered for publication.
Author Response
Here attached Reply to Comments
